# Region-Specific Gut Microbiome Variation Between Changle Geese and Yellow-Feathered Broilers: Correlations with Growth and Intestinal Development

**DOI:** 10.3390/microorganisms13092145

**Published:** 2025-09-13

**Authors:** Dingcheng Ye, Jianxing Qiu, Zitao Fan, Luwei Zhu, Chengyong Lv, Pingting Guo

**Affiliations:** 1Institute of Animal Husbandry and Veterinary Medicine, Fujian Academy of Agricultural Sciences, Fuzhou 350013, China; ydcfafu@126.com; 2College of Animal Science, Fujian Agriculture and Forestry University, Fuzhou 350002, China; q1904442585@163.com (J.Q.); 18022231359@163.com (Z.F.); 12406009010@fafu.edu.cn (L.Z.); l19979258233@163.com (C.L.)

**Keywords:** yellow-feather broiler, Changle goose, intestinal microbiota, body weight, intestinal length

## Abstract

This study comparatively analyzed the spatial heterogeneity of the gut microbiome across gastrointestinal segments in Changle geese versus yellow-feathered broilers to discover their links with growth and intestinal development. Twelve 63-day-old male yellow-feathered broilers and twelve 70-day-old male Changle geese were selected. Body weight (BW), slaughter weight (SW), absolute lengths of the small intestine (LSI) and cecum (LC), and their relative lengths normalized to body size (RLSI/RLC) were measured. Additionally, 16S rDNA sequencing of crop, proventriculus, gizzard, jejunum, cecum, and rectum microbiota was conducted to assess microbial diversity, composition, and its correlation with phenotypes. Results demonstrated higher BW, SW, LSI, LC and lower RLSI and RLC in geese versus broilers (*p* < 0.001). Alpha diversity analysis revealed lower microbial richness and diversity in broilers across most gastrointestinal segments (*p* < 0.05), while beta diversity analysis confirmed distinct community structures between two species (*p* = 0.001). Firmicutes dominated broiler gut microbiota (94.49%), whereas geese exhibited greater phylum-level diversity (*p* < 0.05). Random forestry analysis identified Top 15 core Amplicon Sequencing Variants in both the cecum and rectum, with ASV12260 (unclassified Lachnospiraceae) and ASV12412 (*uncultured Faecalibacterium* sp.) as key biomarkers. Correlation analyses found 21 phenotype-related ASVs (*p* < 0.05). Specially, two *Lactobacillus ingluviei* strains showed negatively correlated with LSI and RLSI in the chicken foregut (*p* < 0.05). And two *Gallibacterium anatis* strains were associated with RLSI, with one strain also showing an inverse correlation with LSI in the goose foregut (*p* < 0.05). Interestingly, one *Peptococcus* strain was negatively correlated with BW and SW, while the other was inversely associated with LC and RLC in the goose hindgut (*p* < 0.05). These findings provide insights into species-specific distribution patterns of gut microbiota across poultry species and their correlation with growth performance and intestinal development, developing a theoretical foundation for advancing avian digestive physiology research and optimizing feeding strategies.

## 1. Introduction

Poultry production is a crucial part of livestock production to gain meats, eggs and other animal products that satisfied human daily nutritional needs [1]. Yellow-feathered broilers (YFBs) are characteristic indigenous broiler varieties of China, including all yellow-feathered meat chickens native to southern regions. The unique flavor and preferable color of meat are overwhelmingly welcomed by consumers in China. Thus, to meet market demands for both volume and quality, studies on the growth performance and intestinal health of YFBs continue to be a high-priority research area [2]. Changle geese, originating from Fujian province in China, are celebrated for their strong adaptability and high crude fiber utilization, playing a significant role in herbivorous poultry farming. However, systematic research on their growth pattern, developmental traits, and digestive physiological characteristics remains relatively insufficient [3].

As the primary site for nutrient digestion and absorption, intestinal length and structural attributes are directly linked to dietary adaptation and metabolic efficiency. Variations in intestinal morphology often reflect evolutionary adaptations to specific dietary niches, influencing both the efficiency of energy extraction and overall physiological performance. For example, after adjusting for body mass, herbivorous avian species demonstrated significantly greater cecal length compared to omnivorous species, with this variation being primarily attributable to dietary classification [4]. Such structural differences are critical for the fermentation of plant-based materials, which requires extended retention times and microbial activity. YFBs (omnivorous) and Changle geese (herbivorous) belonged to the orders Galliformes and Anseriformes, respectively, two phylogenetically distinct groups that have evolved divergent digestive strategies. Their marked differences in dietary habits and metabolic patterns—such as the reliance on fibrous vegetation versus more readily digestible omnivorous diets—likely drove species-specific divergence in growth metrics and intestinal morphology, further underscoring the role of diet in shaping anatomical evolution.

Meanwhile, the gastrointestinal microbiota of poultry plays pivotal roles in nutrient digestion, immune modulation, and overall growth performance. These effects are partly mediated through the production of various metabolites, such as short-chain fatty acids (SCFAs), which are generated via the fermentation of dietary fibers by anaerobic bacteria residing predominantly in the colon [5,6]. The structural and functional composition of the gut microbiota is closely linked to the intestinal microenvironment and the dietary habits of the host. For instance, herbivorous birds typically exhibit a gut microbiome enriched with fiber-degrading bacteria (e.g., Firmicutes and Bacteroidetes), which facilitate the breakdown of complex plant polysaccharides. In contrast, omnivorous species often harbor microbial communities that are more adapted to the catabolism of diverse carbohydrates, proteins, and lipids, reflecting their varied dietary intake [7]. Such differences in microbial ecology not only influence metabolic efficiency but also contribute to host health and disease susceptibility through mechanisms involving microbial-derived metabolites and immune interactions. Nevertheless, current research predominantly focuses on gut microbial characteristics within an individual avian species or microbiota variation across different breeds of the same species [8,9]. There is a lack of systematic study regarding how the gut microbial community structures vary across different intestinal segments between poultry species with vastly different feeding habits. More importantly, the specific correlations between these region-specific microbial communities and the host’s growth performance and intestinal development remain insufficiently discovered, representing a critical knowledge gap [10,11,12].

Therefore, this study focused on YFBs and Changle geese, systematically compared their phenotypic traits such as body weight and small intestinal length, and analyzed the compositional differences in microbial communities across different regions of the digestive tract, to reveal the adaptive divergence in growth performance and intestinal morphology among avian species with different feeding habits, then to provide a theoretical basis for understanding species-specific mechanisms of digestive physiology in poultry and optimizing the feeding and management strategies for local poultry breeds.

## 2. Materials and Methods

### 2.1. Ethics Approval

All animal management and experimental procedures complied with protocols approved by the Fujian Agriculture and Forestry University Animal Care and Use Ethics Committee and were conducted in accordance with relevant guidelines and regulations (Approval ID: PZCASFAFU22038).

### 2.2. Experimental Diets

Feedstuffs for broiler diet were supplied from Fujian Jinhualong Feed Co., Ltd. (Fuzhou, China), while those for goose diet were provided by Fujian Jinzhenghe Feed Co., Ltd. (Fuzhou, China). The basal diet of YFBs formulated according to the feeding standards (NY/T33-2004) is shown in Appendix A and the basal diet of Changle geese designed according to the feeding standard DB32/T 2691-2014 is revealed in Appendix A. All feeds were of plant-based origin and free from any antibiotic additions.

### 2.3. Experimental Animals and Rearing Environment

Twelve 63-day-old male Yellow-Feathered Broilers (YFBs) were randomly chosen from the Poultry Experimental Station of Fujian Agriculture and Forestry University (Fuzhou, China), along with twelve 70-day-old male Changle geese sourced from Changle Goose Breeding Farm (Fuzhou, China). All animals have reached adulthood.

Broilers were raised in cage (1562.5 cm^2^/bird) with 23 h of light, ad libitum feeding and water intake. The temperature of the bird room was controlled at 35 °C for the first week and then lowered by 2 °C to 3 °C per wk until 22 °C.

Geese were reared on elevated wire (Shengfeng Chemical Fiber Rope & Net Co., Ltd., Binzhou, China) floors, which were covered with a plastic mesh (Shengfeng Chemical Fiber Rope & Net Co., Ltd., Binzhou, China) (mesh size: 1.25 cm × 1.25 cm). The flooring was divided into multiple compartments, each measuring 100 cm × 80 cm. A walkway was reserved along one side of the raised floor to facilitate feeding and watering. The walkway was enclosed with soft netting (Shengfeng Chemical Fiber Rope & Net Co., Ltd., Binzhou, China) to prevent the poultry from escaping. The temperature of the bird room was controlled at 28 °C to 32 °C for the first five days and then lowered by 3 °C per week until 21 °C.

The breeding sections, roads, and surrounding environment of the poultry houses were cleansed regularly, and the interior of the poultry houses was disinfected (Weizhenyuan Pharmaceutical Technology Co., Ltd., Sanming, China) on a weekly basis during the rearing period. All poultry were routinely administered the corresponding vaccines (Shengwei Bio-technology Co., Ltd., Nanping, China) to prevent diseases.

### 2.4. Sample Collection

After an overnight fasting period, the body weight (BW) of each individual broiler and goose was accurately measured using a high-precision digital electronic balance (Huazhi Electronic Technology Co., Ltd., Putian, China). Following BW measurement, all birds were humanely euthanized by exsanguination in accordance with ethical guidelines, and the slaughter weight (SW) was immediately recorded.

The entire small intestine, extending from the duodenal origin to the ileocecal junction, was carefully dissected, and all mesenteric attachments were thoroughly removed. The total length was measured and documented as the small intestinal length (LSI). Both paired ceca were isolated from the cecocolic junction to the apical tip, gently flushed with ice-cold 0.9% physiological saline (Labgic Technology Co., Ltd., Fuzhou, China) to clear luminal contents, and the lengths of the left and right ceca were individually measured before summation to calculate total cecal length (LC). Relative intestinal and cecal lengths (RLSI and RLC, respectively) were then normalized against body weight to account for allometric scaling.

Digesta samples from specific segments of the gastrointestinal tract (GIT)—including the crop, proventriculus, gizzard, jejunum, cecum, and rectum—were aseptically collected from each animal. Each sample was promptly transferred into sterile 1.5 mL cryogenic centrifuge tubes (Labgic Technology Co., Ltd., Fuzhou, China), rapidly frozen in liquid nitrogen (Labgic Technology Co., Ltd., Fuzhou, China) to prevent microbial degradation or RNA breakdown, and subsequently stored at −80 °C (Koboku Cold Chain Technology Co., Ltd., Foshan, China) until further processing for microbial DNA extraction and 16S rDNA sequencing.

### 2.5. DNA Extraction and 16S rDNA Sequencing

Detailed methodology for 16S rDNA sequencing closely followed the protocols established in our prior study [13]. The key procedural workflow is summarized as follows. Microbial genomic DNA was extracted from digesta samples using the Stool DNA Kit (D4015-01, Omega Bio-tek, Norcross, GA, USA), strictly in accordance with the manufacturer’s instructions. The V3–V4 hypervariable regions of the bacterial 16S rDNA were then amplified by polymerase chain reaction (PCR) using the universal forward primer F338 and reverse primer R806. The resulting amplicons were purified and quantified to ensure quality and consistency. Sequencing was performed on the Illumina MiSeq Platform (Illumina, San Diego, CA, USA) under standard conditions, adhering to the protocols described in reference [14]. After sequencing, raw reads were subjected to stringent quality control, including filtering of low-quality sequences and removal of chimeras. The remaining high-quality sequences were assembled and analyzed on the Majorbio Cloud Platform (www.Majorbio.com, accessed on 20 April 2023), which provided a comprehensive suite of tools for taxonomic assignment and comparative microbial community analysis.

### 2.6. Sequencing Data Analysis

Alpha diversity analysis at the Amplicon Sequence Variant (ASV) level was performed using Mothur-1.30 software to evaluate microbial community abundance and diversity. This assessment was carried out through the Chao1 index, which reflects species richness, and the Shannon index, which characterizes both richness and evenness of the microbial taxa [15]. To examine the structural variation in microbial communities between geese and chickens, beta diversity analysis was implemented based on Bray–Curtis distance matrices and visualized through Principal Coordinates Analysis (PCoA) using R-3.3.1 with the vegan package [16]. For taxonomic classification, species annotation was conducted in QIIME 2 (v2020.2) by employing a pre-trained Naive Bayes classifier aligned against the SILVA 138 reference database, which contains high-quality 16S bacterial rRNA sequences [17]. Furthermore, the microbial community composition was analyzed at both phylum and genus levels, and the results were graphically summarized using a community Circos plot generated in python-2.7 software to illustrate taxon abundances and relationships [18]. Correlation analyses were performed using the Spearman method to assess associations between phenotypic traits—including body weight (BW), slaughter weight (SW), small intestine length (LSI), and relative small intestine length (RLSI)—and ASV abundances in the foregut regions (crop, proventriculus, gizzard, and jejunum) of both chickens and geese. Similarly, correlations between traits (BW, SW, LC, RLC) and microbial ASVs in hindgut segments (cecum and rectum) were also examined. Finally, random forest analysis, applied without multiple testing correction, was employed to identify the most discriminative feature ASVs within each anatomical section of the cecum and rectum [19].

All raw sequencing data have been deposited in the NCBI Sequence Read Archive under the BioProject PRJNA791802 and PRJNA883760.

### 2.7. Statistics

Data pertaining to phenotypic traits, including body weight, slaughter weight, and various intestinal morphological measurements, were subjected to statistical analysis using the Student’s *t*-test within SPSS 19.0 software (Chicago, IL, USA). All values are expressed as mean ± standard deviation to represent the central tendency and variability of the data. Differences between groups were considered statistically significant when *p*-values were less than 0.05, with specific levels of significance indicated by asterisks as follows: *, *p* < 0.05, **, *p* < 0.01, ***, *p* < 0.001.

## 3. Results

### 3.1. Phenotype Comparison

As evident from Figure 1, geese significantly outperform broilers in BW, SW, LSI, and LC, while underperforming in RLSI and RLC (*p* < 0.001).

### 3.2. Shared and Unique ASVs

1700 ASVs were exclusively recovered from chickens, 4806 ASVs from geese and 292 ASVs were shared between the two species. In the crop, chickens and geese shared 19 ASVs, while harboring 129 and 977 exclusive ASVs, respectively. Within the proventriculus, 58 ASVs were common to both species, with chickens exhibiting 198 unique ASVs and geese possessing 2279 exclusive ASVs. The gizzard microbiota showed 93 shared ASVs, alongside 1103 chicken-specific and 1784 goose-specific ASVs. In the jejunum, 49 ASVs were shared between species, while chickens maintained 224 exclusive ASVs and geese contain 1110 unique ASVs. For the cecum, 52 ASVs were common, with chickens displaying 518 exclusive ASVs and geese 1110 unique ASVs. Within the rectum, 54 ASVs were shared, while chickens and geese possessed 504 and 1026 exclusive ASVs, respectively (Appendix A).

### 3.3. Alpha and Beta Diversity Analysis

As shown in Figure 2, except for the rectum, Chao indexes in each segment of GIT from broilers were significantly lower than geese (*p* < 0.001). The same outcome was observed in Shannon indexes (*p* < 0.05).

PCoA analysis based on Bray_Curtis distance indicated that the microbial structures between FYBs and Changle geese in different parts of GIT were significantly distinctive (R = 0.729, *p* = 0.001) (Figure 3).

### 3.4. Community Composition Analysis

Spatial heterogeneity was observed in both GIT microbial community of chickens and geese at the phylum level. The proportions of Firmicutes, Proteobacteria, Campilobacterota, Bacteroidota and Actinobacteriorta were notably different between chickens and geese (*p* < 0.05). As the most dominant phyla throughout the GIT of both chickens and geese, Firmicutes abundance was richer in chickens (94.49%) than geese (60.23%), while other phyla exhibited higher proportions in geese compared to chickens (Figure 4A).

At the genus level, the remarkable differences in relative abundances between chickens and geese were observed as well (*p* < 0.05). *Lactobacillus* belonging to Firmicutes was the most occupied genus in both the GIT of chickens (62.95%) and geese (14.51%). Different from chickens, the secondary dominant genus in geese GIT was *Helicobacter*, a member of Proteobacteria (Figure 4B).

### 3.5. Core ASVs Analysis

The random forestry analysis revealed Top 15 core ASVs in the cecum and rectum of both YFBs and Changle geese using Area Under Curve (AUC) validation. The core ASVs in the cecum and rectum were predominantly classified as Lachnospiraceae, *Lactobacillus*, *Faecalibacterium*, and *Parabacteroides*. ASV12260 annotated to unclassified Lachnospiraceae and ASV12412 mapped to *uncultured Faecalibacterium* sp. emerged as the most abundant ASVs in the cecum and rectum, respectively (AUC = 1.0) (Appendix A).

### 3.6. Correlation Analysis Between ASVs and Phenotypes

As shown in Figure 5, ASV2487 and ASV2372 annotated to Bacilli and *Lactobacillus* separately were negatively correlated with BW in the chicken foregut (*p* < 0.05). ASV11516, ASV1112, and ASV12636 mapped to *Helicobacter pullorum*, *Lactobacillus*, and *Ruminococcus torques group* in the chicken hindgut and ASV2206 assigned to *Peptococcus* in the goose hindgut were inversely correlated with BW (*p* < 0.05).

ASV2487 in the chicken foregut and ASV2525 annotated to *Lactobacillus aviarius* in the goose foregut were negatively associated with SW (*p* < 0.05). In the chicken hindgut, ASV12412 mapped to *Faecalibacterium*, ASV13192 assigned to *Romboutsia*, and ASV3098 annotated to *Turicibacter* showed positive correlations with SW, while ASV11516 exhibited a negative correlation (*p* < 0.05). Furthermore, ASV11305 (*Helicobacter*), ASV3263 (*Bacillus*), and ASV2251 (*Bacillus*) were negatively correlated with SW in the goose hindgut (*p* < 0.05).

In the chicken foregut, *Lactobacillus ingluviei*-associated ASVs (ASV2270, ASV1357, and ASV3252) were inversely correlated with LSI (*p* < 0.05). Similarly, *Gallibacterium anatis* (ASV1479) showed a negative correlation with LSI (*p* < 0.05). However, *Gallibacterium* (ASV3215), *Neisseria* (ASV1923), and *Bacillus* (ASV3263) exhibited positive correlations with LSI in the goose foregut (*p* < 0.05). As for RLSI, ASV1357 and ASV3252 (annotated as *Lactobacillus ingluviei*) showed significant negative associations with RLSI in chicken (*p* < 0.05). Conversely, in geese, while ASV1399 and ASV1479 (both classified as *Gallibacterium anatis*) exhibited negative correlations with RLSI, ASV2525 demonstrated a significant direct correlation (*p* < 0.05).

LC correlations in the chicken hindgut revealed ASV12964 as positively correlated and ASV11516 as negatively correlated (*p* < 0.05). Moreover, ASV3451 mapped to *Peptococcus* was inversely correlated with LC in the goose hindgut (*p* < 0.05). As for RLC, ASV12636 was directly correlated with RLC, while ASV6004 assigned to *Odoribacter* sp. Marseille-P2698 was inversely correlated with RLC in the chicken hindgut (*p* < 0.05). Similarly, in geese, ASV3451 showed negative association with RLC (*p* < 0.05) (Figure 5).

## 4. Discussion

Indigenous poultry breeds in China, particularly YFBs and Changle geese, represent valuable genetic resources with substantial economic importance. Their marked differences in dietary habits and metabolic patterns likely drove species-specific divergence in growth metrics and intestinal homeostasis. Our results showed that geese exhibited significantly greater BW, SW, LSI, and LC compared to broilers, while displaying significantly lower RLSI and RLC, which demonstrated that geese might achieve enhanced weight gain with less intestinal resource investment while chickens required longer intestine per unit BW to accomplish digestion and absorption [20,21]. These phenotypic differences might reflect geese’s highly efficient utilization of crude fiber. Their GIT likely provided an expanded microbial fermentation chamber and prolonged digesta retention time, enabling highly efficient breakdown of crude fiber—a major component in their diet [11]. In contrast, chicken’s digestive system was reported to be optimized for rapid digestion of nutrient-dense feeds, consistent with its omnivorous adaptation to concentrated feedstuffs [22].

Poultry intestinal microbiome formed dynamic mutualistic systems critical to host biology [23]. The revelation of their microbial structure and diversity could be conducive to deepen the understanding of different digestive mechanisms and provide an insight for learning functions of different microbiota [24]. By examining alpha diversity analysis of GIT separated in six different parts, the microbiota of geese in the foregut and cecum was more abundant and diverse than chickens. This suggests that geese possess a more complex intestinal microbial community with potentially unique and undiscovered functional attributes [25,26]. The beta analysis showed that the microbiota could roughly be divided to two clusters, which were significantly distinctive from each cluster. Such outcome revealed the microbial structure of geese distinctively differed from chickens, which might be due to the feedstuff diversity of waterfowl to some extent, particularly their fiber-rich forage diets that significantly affected microbial diversity [27].

In this study, the most dominant phylum of GIT microbiota in YFBs and Changle geese was Firmicutes. Some research showed that a high abundance of Firmicutes may be positively correlated with feeding efficiency, epithelial development, and barrier integrity [9,28]. But unlike chickens whose gut microbiota were overwhelmingly dominated by Firmicutes, geese demonstrated greater diversity at the phylum level. And Proteobacteria, Campilobacterota, Bacteroidota, and Actinobacteriorta were subsequent predominant phyla in both host species. Although Bacteroidota did not represent the most abundant microbial group in either host, its members are noteworthy for their high efficiency in carbohydrate breakdown, potentially enhancing digestive and absorptive processes in poultry [29]. Consistent with phylum-level findings, *Lactobacillus* (a genus within Firmicutes) was the most abundant genus in the gastrointestinal tract of both chickens and geese. Our findings align with those reported in earlier chicken studies [30,31]. The high abundance of *Lactobacillus* in both host species GIT likely results from its strong adaptation to luminal conditions and the host’s marked preference for this mutualistic symbiont [30,31].

Furthermore, Lachnospiraceae and *Faecalibacterium*, both the core bacteria in GIT of chickens and geese, were predominant in the cecum and rectum, respectively. According to the random forestry analysis, the ASV12260 (unclassified Lachnospiraceae) was the Top 1 feature ASV in the cecum and the ASV12412 (*uncultured Faecalibacterium* sp.) was the Rank 1 feature ASV in the rectum. Lachnospiraceae and *Faecalibacterium* are associated with improved feed conversion rates and body weight gain rates in birds [32]. Studies reported that species in *Faecalibacterium* might contribute to gut health by stimulating the production of beneficial metabolites, especially butyrate and other SCFAs, thereby promoting epithelial well-being [33,34].

Correlation analyses showed that there were 21 phenotype-related ASVs. Among these, 15 ASVs belong to Firmicutes (e.g., *Lactobacillus aviarius*, *Lactobacillus ingluviei*, *Ruminococcus torques group*, *Romboutsia*, *Peptococcus* and *Bacillus subtilis*), 4 to Proteobacteria (e.g., *Gallibacterium anatis* and *Neisseria*), 1 to Bacteroidota (*Odoribacter*), and 1 to Campilobacterota (*Helicobacter pullorum*). Interpretations of their functional roles must consider host species, gut compartment, and strain specificity [35].

*Lactobacillus* strains, widely recognized as probiotics, can be applied to improving growth performance by promoting body weight gain, feed conversion ratio, and feed intake of poultry [36,37,38]. *Lactobacillus aviarius* could secrete glucanotransferase to degrade starch to yield linear ɑ-glucan products (a soluble dietary fiber), and was reported to enhance feed utilization and promote poultry growth [39]. Another strain, *Lactobacillus ingluviei*, is positively correlated with weight gain of ducks [40], but negatively related to LSI and RLSI in the chicken foregut in our study. Similarly, while *Ruminococcus torques group* showed positive correlation with cecal development (RLC)—aligning with reports of its role in bile acid transformation and weight regulation [41,42]—other Firmicutes exhibited species-specific effects. *Romboutsia* sp. correlated positively with SW in YFB hindguts, contradicting its negative link to broiler feed efficiency [43]. Likewise, *Peptococcus* sp. related positively to goose hindgut growth metrics, contrasting broiler studies suggesting improved feed efficiency [44,45]. These discrepancies highlight the importance of host species and gut niche on bacterial function.

ASVs annotated as known pathogenic taxa demonstrated significant alignments with their documented detrimental impacts on host health and productivity. For instance, *Gallibacterium anatis*, a bacterium implicated in systemic infections and reproductive disorders in poultry as well as overall diminished flock performance [46], exhibited significant negative correlations with measures of intestinal development—specifically with both LSI and RLSI in the foregut of geese. Similarly, *Helicobacter pullorum*, which has been widely reported as a causative agent of enteritis, hepatitis, and growth retardation in broiler chickens [47,48], was strongly negatively associated with multiple key growth metrics in the chicken hindgut, including BW, SW, and LC. These inverse correlations are consistent with the well-characterized pathogenic nature of these microorganisms and reinforce their potential role in impairing host physiological development and economic production traits.

There are also some notable host-specific exceptions. The observation that *Neisseria* sp. correlated positively with LSI in Changle geese contrasts with its implication in embryo mortality in wild geese [49]. Similarly, while *Bacillus subtilis* strains reportedly enhance broiler growth and intestinal development [50], no such effect was detected here. This reinforces the context-dependent functionality of gut microbiota and underscores the need for host-adapted probiotic strategies.

## 5. Conclusions

On the whole, Changle geese exhibited higher body weight versus yellow-feathered broilers, with a more diverse and abundant gut microbiome. Furthermore, the findings highlighted the significant association between region-specific gut microbiota (e.g., *Lactobacillus ingluviei*, *Gallibacterium anatis*, and *Peptococcus*) and growth performance as well as gastrointestinal tract development in poultry. Future studies should employ biological analysis approaches to elucidate the functional mechanisms of key microbial taxa and their molecular interactions with the poultry hosts.

## Figures and Tables

**Figure 1 microorganisms-13-02145-f001:**
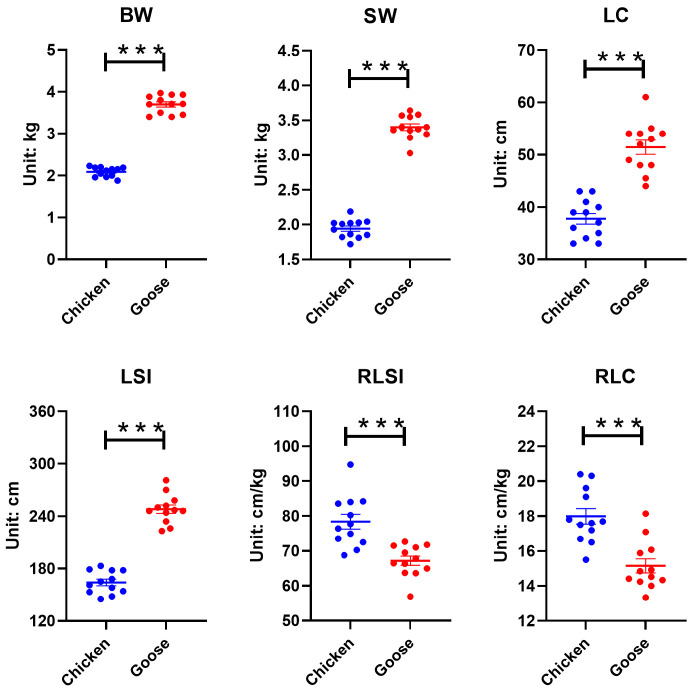
The phenotype comparison between chickens and geese. BW = body weight; SW = slaughter weight; LSI = the length of small intestine; LC = the length of cecum; RLSI = the relative length of small intestine; RLC = the relative length of cecum. ***, *p* < 0.001.

**Figure 2 microorganisms-13-02145-f002:**
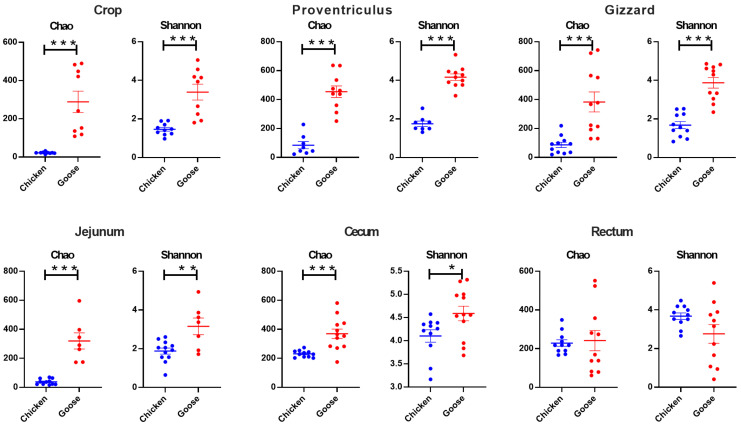
The Alpha diversity analysis of gastrointestinal microbiota in chickens and geese. *, *p* < 0.05, **, *p* < 0.01, ***, *p* < 0.001.

**Figure 3 microorganisms-13-02145-f003:**
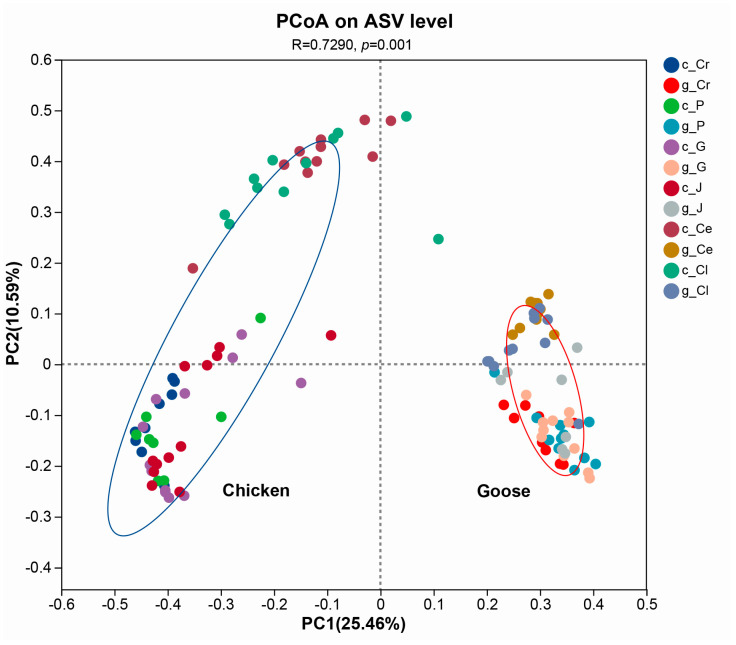
The PCoA analysis of GIT microbiota in chickens and geese. GIT = gastrointestinal tract; Host symbols: c = chicken, g = goose; Intestinal segments: Cr = crop, *p* = proventriculus, G = gizzard, J = jejunum, Ce = cecum, Cl = rectum. Format: Labels follow the pattern [Host]_[Segment] (e.g., c_Cr = chicken crop; g_Ce = goose cecum). R = 0.7290 means the model accounts for the data very effectively, *p* = 0.001 stands for significant result.

**Figure 4 microorganisms-13-02145-f004:**
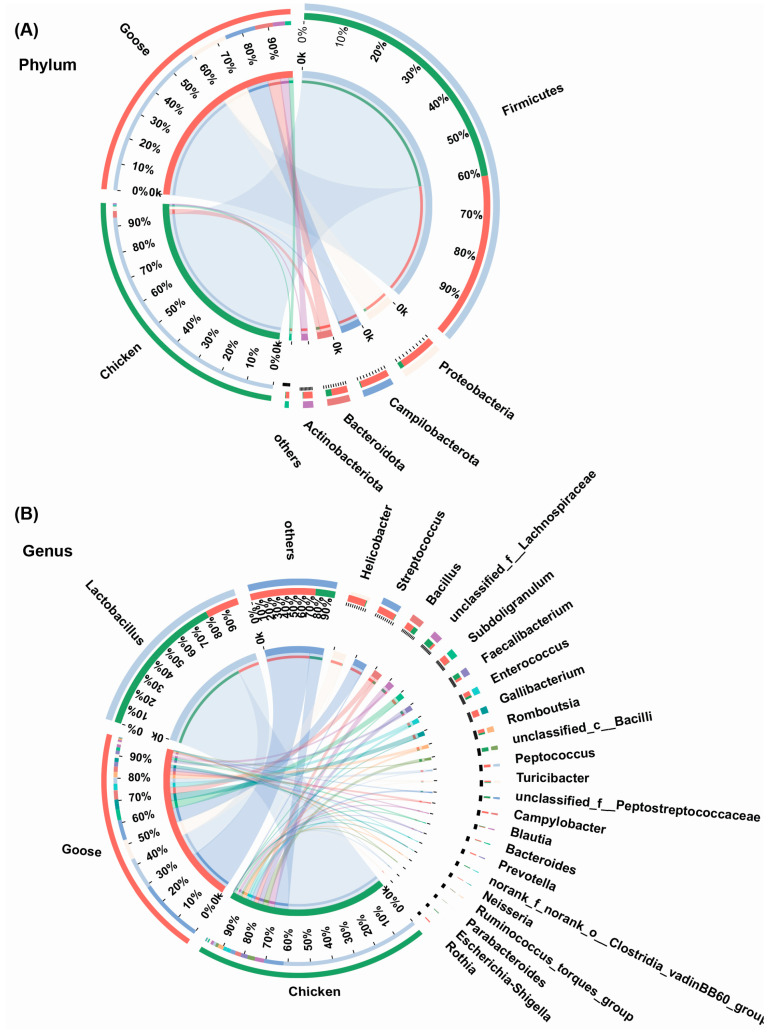
The community composition of chickens and geese gastrointestinal microbiota at phylum (**A**) and genus (**B**) levels. The color of the outer ring corresponds to the label beside it, while the color of the inner ring represents the proportion of its corresponding label within the respective outer ring.

**Figure 5 microorganisms-13-02145-f005:**
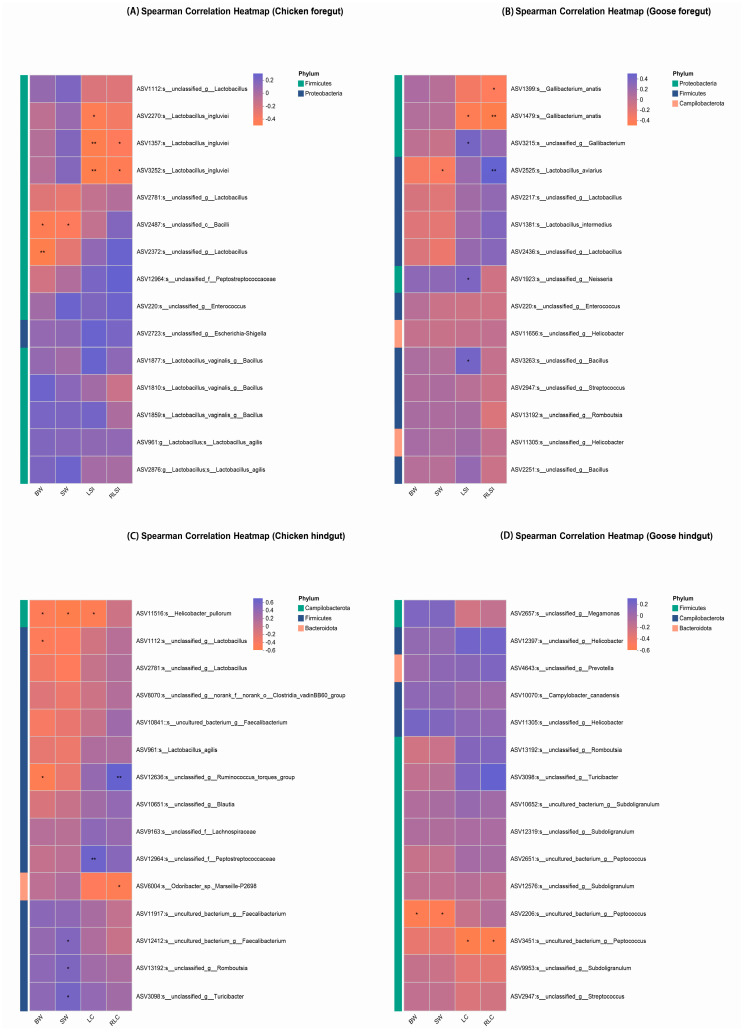
The association analysis between phenotypes and ASVs of chicken foregut (**A**), goose foregut (**B**), chicken hindgut (**C**), and goose hindgut (**D**) microbiota, respectively. BW = body weight; SW = slaughter weight; LSI = small intestine length; RLSI = relative small intestine length; LC = cecal length; RLC = relative cecal length. The foregut was defined as comprising the crop, proventriculus, gizzard, and jejunum. The hindgut was defined as comprising the cecum and rectum. *, *p* < 0.05, **, *p* < 0.01.

## Data Availability

The data presented in this study are available within the article.

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
