# Peer review of "Region-Specific Gut Microbiome Variation Between Changle Geese and Yellow-Feathered Broilers: Correlations with Growth and Intestinal Development"

_microorganisms, 2025, doi:10.3390/microorganisms13092145_

Round 1

Reviewer 1 Report

Comments and Suggestions for Authors

Points Requiring Improvement

Description of rearing conditions. Provide a minimal but clear description of the birds’ rearing environment, including: Type of litter used and whether it was reused. Health management practices (e.g., vaccination schedules, disinfection protocols). Feed composition and ingredients, specifying whether they were animal or plant-based, and indicating the use of antimicrobials... Currently, details differentiating species (diet, management, health history, antimicrobial use) are missing, which limits meaningful comparisons.

Performance data. Report basic performance indicators such as feed intake and body weight, including at least averages and a measure of variability (e.g., standard deviation or standard error).

Sample size and age justification. Provide a rationale for the relatively small sample size and for the choice of bird age used in the study.

Statistical Analyses. The manuscript reports numerous correlations using Spearman and Random Forest methods but does not clearly indicate whether corrections for multiple testing were applied. This needs clarification.

Causality and association. Emphasize that the reported relationships are associative and exploratory. Without functional validation, they should not be interpreted as causal.

Clarity and Presentation. Consider moving extensive tables and/or figures to the supplementary material to improve readability. Summarize the main findings more concisely in the text.

Author Response

Response to the comments:

  1. Description of rearing conditions. Provide a minimal but clear description of the birds' rearing environment, including: Type of litter used and whether it was reused. Health management practices (e.g., vaccination schedules, disinfection protocols). Feed composition and ingredients, specifying whether they were animal or plant-based, and indicating the use of antimicrobials... Currently, details differentiating species (diet, management, health history, antimicrobial use) are missing, which limits meaningful comparisons.

Response: Thanks for your advice. We have added rearing environment and feed standard. (Lines 113-122, 128-143)

  1. Performance data. Report basic performance indicators such as feed intake and body weight, including at least averages and a measure of variability (e.g., standard deviation or standard error).

Response: Thanks for your reminder, but we didn't record the feed intake. Body weights of chickens and geese are represented in Figure 1. Results were presented as mean ± standard deviation.

  1. Sample size and age justification. Provide a rationale for the relatively small sample size and for the choice of bird age used in the study.

Response: The sample size was determined based on the following references. With respect to age, all birds were selected for the experiment upon reaching full physical maturity. (Line 127)

References: Jin, Y.; Guo, Y.; Zheng, C.; Liu, W. Effect of Heat Stress on Ileal Microbial Community of Indigenous Yellow-feather Broilers Based on 16S rRNA Gene Sequencing. Vet Med Sci 2022, 8, 642–653, doi:10.1002/vms3.734. Zhou, H.; Guo, W.; Zhang, T.; Xu, B.; Zhang, D.; Teng, Z.; Tao, D.; Lou, Y.; Gao, Y. Response of Goose Intestinal Microflora to the Source and Level of Dietary Fiber. Poultry Science 2018, 97, 2086–2094, doi:10.3382/ps/pey045. Hiż ewska, L.; Osiak-Wicha, C.; Tomaszewska, E.; MuszyÅ„ski, S.; Dobrowolski, P.; Andres, K.; Schwarz, T.; Arciszewski, M.B. Morphometric Analysis of Developmental Alterations in the Small Intestine of Goose. Animals 2023, 13, 3292, doi:10.3390/ani13203292.

  1. Statistical Analyses. The manuscript reports numerous correlations using Spearman and Random Forest methods but does not clearly indicate whether corrections for multiple testing were applied. This needs clarification.

Response: Thanks for your suggestion. Multiple testing correction controls the risk of a surge in false positives caused by testing a large number of features at once, thereby ensuring the reliability of conclusions. However, since the sample size in our study is relatively small, multiple testing correction is not necessary. (Line 209)

  1. Causality and association. Emphasize that the reported relationships are associative and exploratory. Without functional validation, they should not be interpreted as causal.

Response: Done as requested. (Line 334, 337-338, 341, 350-352, 360-362, 398, 409)

  1. Clarity and Presentation. Consider moving extensive tables and/or figures to the supplementary material to improve readability. Summarize the main findings more concisely in the text.

Response: Thanks for your advice, and we have moved Figure 2 and Figure 6 to supplementary figures and modified previous Figure 1 and Figure 3 to improve readability.

Reviewer 2 Report

Comments and Suggestions for Authors

The abstract does not provide the number of animals 

In introduction add more recent references 

Highlight the gap in research field and objective of the study 

In discussion, strengthen the interpretation by linking your results to the underlying physiological mechanisms

Lines 284-296: There is only one reference. Add more references

Lines 326-332 : Add references 

Lines 306-310: Rewrite the sentence more clear

Lines 306, 315, 343: Ensure that all scientific names are written in italics

In conclusion suggest future directions in research field 

Author Response

Response to the comments:

  1. The abstract does not provide the number of animals

Response: Done as requested. (Line 16-17)

  1. In introduction add more recent references

Response: Thanks for your advice. We have added three recent references in Introduction. (Line 96)

  1. Highlight the gap in research field and objective of the study

Response: Thanks for your reminder. We have highlighted the gap in research field and objective of the study. (Line 90-105)

  1. In discussion, strengthen the interpretation by linking your results to the underlying physiological mechanisms

Response: Done as requested. (Line 337-338, 350-352, 366-372, 410-423)

  1. Lines 284-296: There is only one reference. Add more references.

Response: Thanks for your reminder, we have added two references. (Line 348, 352)

  1. Lines 326-332: Add references.

Response: Thanks for your suggestion. We have added one reference. (Lines 393)

  1. Lines 306-310: Rewrite the sentence more clear.

Response: Done as requested. (Lines 364-372)

  1. Lines 306, 315, 343: Ensure that all scientific names are written in italics.

Response: Thanks for your reminder. However, the papers cited below indicate that, according to standard convention, only bacterial genus and species names should be capitalized.

References: Zhang, X.; Akhtar, M.; Chen, Y.; Ma, Z.; Liang, Y.; Shi, D.; Cheng, R.; Cui, L.; Hu, Y.; Nafady, A.A.; et al. Chicken Jejunal Microbiota Improves Growth Performance by Mitigating Intestinal Inflammation. Microbiome 2022, 10, 107, doi:10.1186/s40168-022- 01299-8. Kobayashi, R.; Nagaoka, K.; Nishimura, N.; Koike, S.; Takahashi, E.; Niimi, K.; Murase, H.; Kinjo, T.; Tsukahara, T.; Inoue, R. Comparison of the Fecal Microbiota of Two Monogastric Herbivorous and Five Omnivorous Mammals. Animal Science Journal 2020, 91, e13366, doi:10.1111/asj.13366. Yao, T.; Wang, C.; Liang, L.; Xiang, X.; Zhou, H.; Zhou, W.; Hou, R.; Wang, T.; He, L.; Bin, S.; et al. Effects of Fermented Sweet Potato Residue on Nutrient Digestibility, Meat Quality, and Intestinal Microbes in Broilers. Animal Nutrition 2024, 17, 75 – 86, doi:10.1016/j.aninu.2024.03.007.

  1. In conclusion suggest future directions in research field.

Response: Done as requested. (Line 438-440)

Round 2

Reviewer 1 Report

Comments and Suggestions for Authors

I agree.